# Visions of Red Riding Hood: Transformative Bodies in Contemporary Adaptations

Elizabeth Abele

Gulf University for Science & Technology, Hawally 32093, Kuwait; abele.e@gust.edu.kw

**Abstract:** Gothic and sexual elements are embedded within both Charles Perrault's and the Brothers Grimm's tellings of "Little Red Riding Hood". When popular culture turned to fairy tales from the late 20th century forward, reimagining them as gothic tales for adults, "Little Red Riding Hood" provided a particularly rich setting. In particular, these adaptations exploited the false binaries within these tales while making more visible the sexual abuse and recovery encoded in the narratives. This essay will first explore the particular gothic qualities within this tale, as well as the shapeshifting nature of the four characters. After establishing how the figure of Red, as well as her motifs, are key to ensemble fairy-tale narratives, I will examine adaptations that directly explore the sexuality and agency of a young woman, as she resists both predators and her family legacy. However, the last section will note that monstrosity, like victimization, can be resisted. Overall, this essay interrogates contemporary film and television adaptations of this tale, with a particular interest in the messages of recovery and agency in these new versions.

**Keywords:** fairy tale; gothic; film adaptation; feminist; sexual abuse

## 1. Introduction

My mapping of Red Riding Hood's journey through the forest began in the classroom in 2006. I designed my Film and Literature course for students to examine how genres and issues develop differently between the two media and over time. Fairy tales proved a useful pedagogical tool for students to master several foundational theories of the gothic, adaptation, and gender. Coincidentally, while I was teaching this course, gothic revisionings of fairy tales became very common in Hollywood and television. "Little Red Riding Hood" emerged as a particularly productive class text because of the tension between the print versions, classic and contemporary, in their portrayals of a young girl's treacherous journey to womanhood.

The study of Red Riding Hood and her Wolf is complicated from the start by the two "originals": Charles Perrault's "Little Red Riding Hood" published in 1697, followed by The Brothers Grimm's "Little Red Cap" in 1812[1] (both in turn derived from conflicting folk sources). My students recognized that both the Brothers Grimm's "Little Red Cap" and Charles Perrault's "Little Red Riding Hood" differed from their childhood's brightly colored picture books or cartoons, which revolved around a girl who was definitely pre-teen. The overt moral of Perrault's "Little Red Riding Hood" forced them to reconsider the sexualities of both Red and the Wolf: "pretty girls with charm,/Do wrong and often come to harm . . . /if they don't do as they ought, /It's no surprise that some are caught/By wolves who take them off to eat" (2010, p. 162). Perrault is explicit in placing the blame for the attack on "pretty girls with charm" who do not do as they "ought".

Jacob and Wilhelm Grimm's "Little Red Cap" more subtly portrays the Wolf's attack as a sexual encounter. However, the figure of the Woodsman introduces a violent and gory scene: his cutting open the wolf's belly, with Red and Grandmother emerging bloody. This is followed by an encounter with a second wolf, which Grandmother tricks into a pot of boiling water. The sexuality and violence had been removed from my students' childhood

versions, leading to their realizations that they had grown up with an "adaptation". As Andrew (2000) advocates, my students learned to see adaptation as a "complex interchange between eras, styles, nations, and subjects" (p. 36). As their childhood versions reflected the different ideas 20th-century parents had about tales acceptable for children, the late 20th-century version of the tales connected to more contemporary attitudes toward female sexuality and agency—traces of which can be found in the Perrault and Grimms' tales.

These shapeshifting qualities, tied to dynamics of sexual development and desire, human and animal, predator and prey, drive the late 20th-century and 21st-century adaptations of "Little Red Riding Hood". More gothic and feminist versions of the tale began in print with Angela Carter's "Company of Wolves" (1979), Roald Dahl's "Little Red Riding Hood and the Wolf" (1982), and Tanith Lee's "Wolfland" (1983); violence comes more naturally to the female protagonists, bordering on bloodlust. This lady in Red then appeared on screen as a key figure in the ensemble fairy tale films *The Brothers Grimm* (Gilliam 2005) and *Into the Woods* (Marshall 2014), before qualifying for direct adaptations in *Freeway* (Bright 1996) and *Red Riding Hood* (Hardwicke 2011), or intertextually in *Woodsman* (Kassel 2004). Together these works established the potential of this girl and her supporting characters.

This essay will examine contemporary Hollywood revisionings of "Little Red Riding Hood" as central to onscreen feminist and gothic restagings of European folk tales in the last decades. This tale of this humble girl, caught between childhood and womanhood, provides a useful figure for adaptations to craft new fables for adults. Together, these adaptations of Red across media build on the gothic subtexts of this children's tale of sexual awakening, predators, and family legacies, as well as the potential for female agency and forbidden desires—raising timeless and timely questions about overcoming victimhood and monstrosity. These film adaptations foreground the mutability and possibilities embedded in "Little Red Riding Hood", a gothic richness that previous generations preferred to ignore, choosing repression over potentiality.

## 2. Following Red's Gothic Trail

In their concern for innocent maidens in peril, European fairy tales anticipated the generic structure of the 18th-century gothic novel, though setting the action in wooded spaces rather than crumbling estates. Embedded within these ecogothic spaces are found the gothic qualities of doubling, extreme good and evil, old and young, past sins, and repressed desires—with a resolution that restores moral order. In the past 50 years, popular culture has exploited these gothic elements, returning to the Grimm-er source to create gothic fiction, films, and TV series—geared to an adult audience. These adaptations still present the tension of a young girl in the process of becoming a woman, as well as the ability of a wolf (two- or four-legged) to pretend to be a friend or a grandmother.

While Perrault and the Grimms may have presented a girl who was "sweet", contemporary writers such as Carter and Lee generally found that characterization too limiting, setting the stage for Hollywood's revisioning of Red, who may not be little or good. Later, "Little Red Riding Hood" provided particularly fruitful soil for Hollywood to examine the treacherous territory of female and male relationships. Zipes (2012) notes its unique position as "the most popular memetic tale that enunciates the gender conflict and alludes to the problem of rape" (p. 147). Significantly, none of the characters in the original tale have stable shapes: the pubescent Red; the disappearing/reappearing Grandmother; the manlike and transvestite Wolf; and the convenient Woodsman.

This mutability and indeterminacy make "Little Red Riding Hood" particularly appropriate for gothic exploration. Though their source tales may have characterized the predator as a werewolf, Perrault and the Grimms avoid the explicit connection. Bacchilega (1999) sees the added gothic dimension of the werewolf, as any "neighbor or relative [male or female] could be such a shape-shifter and turn dangerous" (p. 55). Without a witch or a prince, both the rescuer and the threat are more ordinary and less visible. Halberstam (1995) describes the gothic monster's relation to anxieties about class, gender, sexuality, and race.

If the gothic monster's body is "remarkably mobile, permeable, and infinitely interpretable" (p. 21), the bodies of "Little Red Riding Hood" provide a rich kaleidoscope.

Another shape-shifting aspect of Red Riding Hood is her age. While 20th-century storybooks pictured her as "Little", this modifier is at odds with Perrault's moral, as well as the coded messages of the Grimms' version. Vaz da Silva (2016) explains that Perrault altered the symbolism of his source material, which had set the protagonist's age closer to 15 years old (p. 174), firmly on the other side of childhood. Bettelheim (1989) notes the grandmother's gift of the red cap symbolizes the transfer of sexual attractiveness, though she may be too young "for managing what this red cap symbolizes, and what her wearing it invites" (p. 173). Though Gretel is a child, and Cinderella and Snow White are prospective wives, Red exists between the two—more than a girl but not quite a bride.

Overall, with the move away from magic and royalty, this gothic tale leaves the confines of a "fairy tale", offering a more accessible coming-of-age narrative. Hogle (2002) describes the tensions central to the gothic: "Threats and longing for gender-crossing, homosexuality or bisexuality, racial mixture, class fluidity, the child in the adult, timeless timeliness, and simultaneous evolution and devolution . . . all these motifs, as possibly evil *and* desirable, circulate through Gothic works across the whole history of the form" (p. 12). These gothic motifs—as well as the tension between evil and desirable—are more visible in "Little Red Riding Hood" than in other fairy tales. In addition, the fluidity and longings exist in the literal conflation of the bodies of the Wolf(man), Red, and Grandmother. This fluidity is then interrupted by the Woodsman, who violently restores order—yet his sudden appearance and disappearance raise additional questions. The symbols and motifs of "Little Red Riding Hood" have made it a rich fable for adult audiences, leading to a range of independent and commercial screen productions.

### 3. Flashes of Red: Referencing the Tale

Overall, contemporary film adaptations of "Little Red Riding Hood" follow Hopkins' (2005) requirement of "Gothicizing" in their focus on polarities, which invest the Gothic's "uncanny ability to hold its darkly shadowed mirrors up to its own age" (p. xi), with the "blurring of previously secure polarities" (p. xi). In adapting this tale, these adaptations "are caught up in the ongoing whirl of intertextual reference and transformation, of texts generating other texts in an endless process of recycling, transformation, and transmutation, with no clear point of origin" (Stam 2000, p. 66). Having escaped Disneyfication, "Little Red Riding Hood" has been particularly amenable to integration into a wide range of period and contemporary texts—particularly as a tale literally of "transformation and transmutation".

While *The Brothers Grimm* and *Into the Woods* employ the full range of the tale's motifs, Red's journey through the woods is particularly central to these ensemble fairy tales—with the woods themselves integral to the action of both films. Parker and Poland (2019) describe the function of the woods in the gothic imagination: "The thought of being alone in these wild, more-than-human spaces still provokes a sense of unease for reasons that go beyond simple physical safety. In the cultural imagination, Nature has always engendered fear, wonder, and fascination" (p. 1). In *The Brothers Grimm*, the woods frame and hide the evil forces targeting the maidens of Marbaden.

Though Red's actual appearance in *The Brothers Grimm* is brief, her presence sets up the complicated nature of the threat. The film focuses on the disappearance of young girls from the village of Marbaden in the neighboring woods. Though the girl in the red cape is not the first missing girl, hers is the first abduction shown. It is not just her journey through the woods that is ominous: the Marbaden Woods themselves literally come alive to confuse and trap trespassers. Red's abduction also introduces the Wolf: after she pricks her finger on berries (bleeding), the unseen Wolf is heard to howl and then chase her. Only her wind-borne cape escapes the woods, announcing her transformation.

*The Brothers Grimm* directly delineates between supernatural enchantment and merely charming predators. When they were boys, Jacob (Heath Ledger) was fooled by a fake magician, leading to their sister's death. As with other former prey that become predators,

this event justifies for Will (Matt Damon) their work as charlatans, ridding villages of "enchantments"—that they had staged. Will's reluctant partner Jake is the scholar who yearns to find actual magic. Will has been a "wolf", using his charm to fleece villagers and to seduce women; early in the film, he is discovered in bed with two women, wearing one of their hats (echoing the Wolf wearing grandmother's cap). Threatened with execution, the brothers must solve the mystery of Marbaden—assumed by the occupying French army to have been staged by conmen like the Grimms.

These Woods become the testing ground of the (previously) charlatan Grimm brothers. As Botting (2017) establishes, "Landscapes stress isolation and wilderness, evoking vulnerability, exposure, and insecurity . . . Nature appears hostile, untamed and threatening: darkness, obscurity, and barely contained negative energy reinforce atmospheres of disorientation and fear" (p. 4). While the Marbaden Woods defeats the French soldiers, the Italian captain and the Grimms eventually succeed within them. Being disoriented and then confronting their vulnerability is what ultimately allows them to escape their previous limitations, freeing them to accept the magical reality of the woods and discover their own potential. The motifs of "Little Red Riding Hood" continue to stitch this film together, signaling unstable bodies and personas. After the Grimms' arrival in Marbaden, the tale is quoted by a young girl (Veronika Loulova), who remarks to a possessed horse: "What big teeth you have". This horse then swallows her whole (like the Wolf). Before galloping away, the horse opens his mouth, showing the girl screaming in its belly; the horse/girl disappears into the Marbaden Woods. Gilliam's increasingly horrific stagings of the girls' abductions resonate with contemporary media coverage and public fears, as "our collective response to dead or 'disappeared' children carries Gothic resonances" (Armitt 2011, p. 18).

Of course, the Wolf/Man is another unstable body that masked his identity: he is a Marbaden villager, a Trapper enchanted by the Mirror Queen (Monica Bellucci). In exchange for saving his life, he must serve her, collecting the 12 girls that she needs to restore her body. Five-hundred years ago, the Queen bargained for eternal life to escape the plague; however, in a curse-like fashion, her beauty is only preserved in her mirror. The enslaved Wolf/Man places each girl in a coffin, drawing blood with a finger prick (symbolizing their fertility). While in the Grimms' tale, Red is reborn via Caesarean, the Marbaden maidens emerge through underground streams. The Wolf/Man serves as both predator and midwife for his Queen, a gothic monster that, as Halberstam argues, cannot be read singularly (21): wolf/man, predator/victim, dead/alive, and lover/eunuch.

The Mirror Queen reverses the transfer of fertility from grandmother to Red: to transform from mummy to beauty, the Mirror Queen must steal the blood of 12 girls at the blood moon. Vaz da Silva (2016) notes that the symbols of "Little Red Riding Hood" link the budding of Red's womanhood with the withering of her grandmother's, since "all the desired things in life, including health and fertility, exist in a finite quality", with older women transmitting their fertility to young women (the gift of the red cape) (p. 183). The Mirror Queen goes against this natural transfer, taking the girls' blood to restore her "moon time".

Despite the diverse fairy tale references, it is the themes of "Little Red Riding Hood" that drive *The Brothers Grimm*. In addition to the waxing/waning of fertility and its unstable bodies, predation also drives the action. While this Wolf/Man is more a panderer than a sexual predator, his role in the collection of the Marbaden girls remains unsettling—particularly since the first two that he takes are his own daughters. However, the film ends with the ultimate transformation, a redemption; for once, the brothers are actual heroes, battling real magic. Jake can now put Will on the appropriate road, as a protector and chronicler instead of a predator.

Red as well as her woods are more central to the fairy tale ensemble of *Into the Woods*.[2] In his film adaptation of the Stephen Sondheim and James Lapine musical, Marshall was able to fully present the complexities of the Woods—a place that not only evokes a "sense of unease" but creates a liminal space that offers the characters the freedom to step outside of their stereotypical positions. Though these Woods may still be a place of fear, it is "wonder

and fascination" (Parker and Poland 2019, p. 1) which may be ultimately of more value to the characters' developments. The refrain of *Into the Woods'* opening song announces that its action will take place away from the village and the castle. The most important action occurs as the major characters—the Baker, the Baker's Wife, Jack, Cinderella, and Little Red Riding Hood—all move "into the woods", individually becoming "different in the woods" as they enter this liminal space. Significantly, it is the Little Girl who is the first to move into the woods, and to describe its dynamics: "I have no fear/Nor no one should . . . . And who can tell/What's waiting on the journey?"

Little Red Riding Hood (Lilla Crawford) is introduced in the musical "Prologue" as a "hungry little girl", emphasizing both her childlike behavior and her appetites (in the film, she is only called "Little Girl" or "the girl"). She fills her arms with goods from the baker, saying she is taking them to her grandmother, as she stuffs her mouth. She admits her lack of concern: "To Granny who/Is sick in bed/ . . . For all that I know/She's already dead". Zipes (2012) explains why Red is generally not a sympathetic victim: "She is more of a wimp. Either she stupidly agrees to an assignment with a hungry wolf and thus is complicit in her own rape/violation and death (Perrault), or she needs the help of a hunter to free her from the wolf's belly (Grimm)" (p. 143). *Into the Woods* goes further than Perrault in justifying Little Red Riding Hood's abuse, presenting her as a tween who uses the excuse of her grandmother to fill her own belly with pastries, foreshadowing her fate in the Wolf's belly.

The sexual abuse is more direct than in *The Brothers Grimm,* though still largely allegorical. While the Wolf (Johnny Depp) addresses her as "Little Girl", his language describes her as both a meal and a sexual opportunity: "scrumptious carnality".[3] The virgin and the old woman are both desirable to this Wolf: "Grandmother first, then Miss Plump/What a delectable couple:/Utter perfection, one brittle, one supple". They combine in his belly to represent the full spectrum of female fertility. After the Wolf carries through on his threat, the Baker (James Corden) conveniently arrives for the rescue, maintaining the female characters' passivity—while demonstrating that a Baker can use a knife.

Yet this version acknowledges Red Riding Hood's loss of innocence. She sings of her transformation: "I know things now/Many valuable things/That I hadn't known before". Though it seems that Little Red Riding Hood has learned her lesson about talking to strangers, she sees the ambivalence of no longer being a little girl: "Isn't it nice to know a lot!/And a little bit not". Though this shift in her position is not addressed again, Beckett (2002) notes how it is physically marked, as she "reappears wearing a fashionable new wolf cape that she protects from theft with a knife" (p. 128).[4] However, though her maturity does not follow a straight line, she gradually becomes thoughtful and considerate, more than just a hungry girl.

Abuse survivor may be another hidden identity within Grimms' version of the tale. O'Malley (2022) observes that abuse history is rarely acknowledged as an identity "but among the parallels to be made with these other markers is abuse's relationship to the phenomenon of passing". Therefore, expanded versions of this tale include not only the Wolf's passing as friend and grandmother but Red's passing as "normal" after her encounter with the Wolf. The wolf-skin coat is a cover for the Little Girl's new position as deflowered. Bettelheim (1989) might further read the coat as symbolizing her connection to the Wolf, an "externalization of the badness a child feels when he (sic) . . . permits himself to tempt, or to be tempted sexually" (p. 177).

Instead of downplaying the gothic qualities of fairy tales, *The Brothers Grimm* and *Into the Woods* put young girls in real danger, reestablishing "fairy stories which have traditionally documented the terrifying ordeal childhood can be" (Armitt 2011, p. 20). Yet on the other side of the ordeals is the potential for growth. Red Riding Hood's real transformation occurs not when she emerges from the Wolf but when she is willing to be a caring member of a family, rather than just the receiver of pastries and cloaks. Similar to Will and Jake, she emerges from the woods a better person. In the mix of other fairy tale

references, these films show the particular resonance of this girl, shaped by her encounter with a wolf.

## 4. Red as Action Heroine

An issue with these adaptations, going back to Carter and Lee, is that they reinscribe the confines of the original tale. As Bacchilega (1999) notes, "devoured or domesticated, charged with sin or in charge of the feminine hearth" (p. 58), Red ends the narrative confined. Happily, film and television adaptations focused directly on Red have each allowed her to escape these limits, even while more directly acknowledging her precarious sexual position.

The independent film *Freeway* (Bright 1996) slightly predates Hollywood's interest in fairy tales. The film follows Perrault's allegory of the predator and young girl, jettisoning the literal wolf as well as the woods' setting. Vanessa (Reese Witherspoon) is the teenage daughter of a crack whore, who has been sexually abused by her stepfather. She takes the California freeway to her estranged grandmother's house to avoid another foster home placement whose dangers she knows. Her connection to the tale is signaled by her red leather jacket and basket, as well as her boyfriend Chopper (Bokeem Woodbine), her Woodsman; he lends her his gun for her trip—aiding her self-protection rather than promising rescue.

Fulfilling Perrault (2010)'s observation that some wolves are "not as friendly as they might appear" but are in fact "the most dangerous wolves of all" (p. 162), Bob Wolverton (Kiefer Sutherland) picks Vanessa up when her car breaks down. Bob seduces her not sexually but with kindness and a hot dinner. His disguise is that of an adolescent counselor; he recognizes the signs of her sexual abuse, which he convinces her to share with him—truths that she has not even shared with Chopper. In his exploration of film portrayals of sexual abuse, O'Malley (2022) notes the marking of "the sexually abused—either explicitly or implicitly—as nonnormative", explaining Vanessa's instinct to pass.

Likewise, Bob is hiding his nonnormative identity: he is actually the I-5 Killer, who kills and rapes girls like Vanessa, calling them trash. As a predator, Bob "smelled" the truth before her confession of abuse. However, Red's past has made her more resilient and resourceful, and she turns Chopper's gun on Bob. Unfortunately, the severely injured Bob rises from the dead and makes it to a hospital, where he convincingly plays the victim. Vanessa's dissociative walk into a diner for pie—while covered in Bob's blood—makes his version of events more convincing. Hogle (2002) notes the power of these gothic and gruesome bodies, allowing the audience "to define ourselves against these uncanny abjections while also feeling attracted to them" (p. 17). Vanessa is both a figure of horror and courage. When she learns that her boyfriend was killed in a drive-by, Vanessa further devolves, with her transformation only exacerbating the system's preconceived notions about her.

Crucially, the detectives are the ones who next change. As they investigate Vanessa, they find the "good" girl hidden underneath what she has done to survive, as she "perform[s] acts of social stealth that range from benign to survivalistic to horrifically nefarious" (O'Malley 2022). The detectives uncover the Vanessa who is a faithful friend, forgiving daughter, and devoted girlfriend. Only then do they take the time to look beyond Bob's disguise, finding the Wolf in Wolverton.

Director Catherine Hardwick moved from the teen vampire-romance *Twilight* (2008) to the full-length *Red Riding Hood* (2011)—dropping the "little" as young woman Valerie (Amanda Seyfried) chooses between two suitors, at the same time that her village is stalked by a werewolf. *Red Riding Hood* moves almost 1000 years earlier than *Freeway* to stage its tale in the village of Daggerhorn; however, it still features a protagonist aware of her own sexuality and agency. Since Valerie's father Cesaire (Billy Burke) is a woodcutter, she and her older sister Lucie (Alexandria Maillot) were raised at the edge of the woods, away from the village—where she has done forbidden things, such as killing a rabbit. Here, she has grown up with the threat of the Wolf, a beast that attacks during the full moon. As

Hogle (2002) describes, gothic texts "project modern concerns into a deliberately vague, even fictionalized past" (p. 16). Though *Red Riding Hood* is not an allegory, it still provides a vague space to explore issues of family legacies and female agency.

The threat of the Wolf is connected to Valerie's sexual maturity, as her forced engagement to the prosperous blacksmith's son coincides with the blood moon[5]—a lunar event that not only increases the Wolf's power but curses anyone bitten at this time to likewise transform. However, her heart belongs to the woodcutter Peter (Shiloh Fernandez). With this engagement, her mother (Virginia Madsen) works to separate Valerie from Peter and the woods. The Wolf's first victim is Lucie; the second is the father of Valerie's betrothed Henry (Max Irons). While *Into the Woods* has a stylishly dressed Wolf on two legs, and *Freeway* a man with wolf-like qualities, *Red Riding Hood* follows Angela Carter and Tanith Lee by resolving the Wolf/Man into a werewolf—a supernatural being that shifts between the two forms. The figure of the werewolf adds a gothic unease since it could be anyone in the village, male or female, while blurring the polarity of man and beast. While Perrault's tale clearly focuses on the Wolf as a sexual predator, this werewolf is solely feared as a killer.

However, the Wolf's particular connection to Valerie appears more sexual than carnivorous. At a dance celebrating the killing of a normal wolf, where Valerie has celebrated her own sexuality, the Wolf arrives, "speaking" to Valerie. His speech is intimate: "You understand me; that is all that matters . . . Let me take you away. You and I are the same". This statement of desire focuses her suspicion on Peter or Henry, causing her to distrust them; in other adaptations, the protagonist Red similarly takes Perrault's moral too much to heart, closing herself not only to wolves but good men.

Yet Valerie's sexual desires are the first red herring. The sexual and monstrous secrets that are behind the blood moon are within her own family. Also suspected is Valerie's Grandmother (Julie Christie): Henry recognizes the scent of the Wolf on her. However, she is merely the unknowing wife and mother of wolves.[6] The current Wolf is Cesaire, who could not leave the village for larger hunting grounds because he "loved" his daughters. The lesser family secret is the infidelity that led to the birth of Lucie. Cesaire kills both Lucie and his mother for their "betrayal". Though Valerie initially is suspicious of all the men in her life, the film ends by showing the possibility both of good men and redemption. Henry willingly steps aside while remaining protective of her. Peter and Valerie together end this Wolf but not the cycle, as Peter is bitten.

These Reds must all come to terms with their family histories and their own potential. Hogle (2002) notes that gothic texts "raise the possibility that all 'abnormalities' we would divorce from ourselves are a part of ourselves, deeply and pervasively (hence frighteningly), even when they provide quasi-antiquated methods to help us place such 'deviations' at a definite, though haunting, distance from us" (p. 12). These young women all exist outside of definitions of "normal" girls, yet their ultimate strength comes from fully knowing who they are. Vanessa and Valerie break away from their family legacies, standing up for themselves and others—by whatever means necessary. These films embrace the gothic potential, allowing these female protagonists to be more than devoured or domesticated.

## 5. Reforming the Wolf

The most interesting point in *Red Riding Hood* is the question raised at the end. Just because Peter is a wolf, does he have to be a monster? Though Valerie does not transform, what qualities does she inherit from her father? Peter leaves the village and Valerie so that he can learn to control his new nature. As she vows to wait for him, she believes that it is possible for them to control their hidden selves. This confrontation with the monster has led to the most thoughtful adaptations of "Little Red Riding Hood". This challenge of resisting a bestial nature is the focus of this provocative independent film *Woodsman* (2004).[7]

In *Woodsman*, Walter (Kevin Bacon) is the ultimate monster—a pedophile who molested girls 10–12 years old. He is adamant that he never hurt them, and that he always asked their age. But having served 12 years in prison, he is aware that his previous criteria are not sufficient for the law, society, or his estranged sister. Rather than adapting "Little

Red Riding Hood", this film has an intertextual relationship to the tale, incorporating its motifs. Walter works at a lumberyard, making furniture as a hobby; to his community, he looked like a Woodsman until his arrest as a Wolf.

The film follows Walter's journey to avoid becoming a monster again. He tells his court-mandated psychologist that he wants to be "normal", and that he would like to wake up one day and not have "thoughts" when he sees a girl. His apartment is across from a school, yet the required distance away from it. As he watches the boy enter and leave the playground, he senses the presence of another predator, whom he dubs Candy (Kevin Rice). The man uses candy to tempt the "cherubs". Walter tells himself, if boys go with him, "they must want to", avoiding the responsibility for what he is witnessing. Instead of reporting the danger, he relates to Candy.

His first step toward normalcy is his relationship with a coworker, Vicki (Kyra Sedgwick). When he confesses his past, that he has been a monster, she does not run away. Instead, she shares that she was "poked" by each of her older brothers and that Walter is the first person she has told: "I'm trying to tell you who I am, if you're interested". With Walter, she chooses to claim her identity rather than pass for "normal". By empathizing with Vicki, as well as witnessing her forgiveness for her "good" brothers, he begins to hope he can be "good". Walter is fully aware that "most people" believe that he cannot change. Vicki believes in the potential of his better nature to emerge.

Representing "most people" is Sgt. Lucas (Mos Def), who regularly calls on Walter. He sees his harassment as justified, for the new acts that Walter is statistically likely to commit. Eventually, he asks Walter if he believes in fairy tales, "the one with the Woodsman". Telling Walter of horrific crime scenes he has witnessed, he laments, "there are no fucking woodsmen in the world", directly referencing the need for Woodsman in the world to kill Wolves (like Walter).

Unfortunately, Walter is tempted by a girl he sees on the bus. Further signaling the film's intertextual relationship to "Little Red Riding Hood", she wears a red coat and is named Robin. Similar to Bob, he kindly talks to this lonely girl. When he asks her to sit on his lap, she says no, sharing that her father sometimes asks her to sit on his lap (like Vicki, Robin has been passing). As he questions her, he realizes not only that her father has molested her (as he had hoped to), but more importantly, he realizes that her father's acts have "hurt" her. Walter can no longer deny that he has been a monster—just because they went with him does not mean they wanted what happened. Choosing now to be a Woodsman, he severely beats Candy after he returns a boy to the playground. As Walter moves in with Vicki, he commits to leaving his Wolf days behind. Though he may not be a vigilante, he can be a watchful Woodsman—this is his version of normal.

This film joins the examination of the most complicated dynamics of "Little Red Riding Hood", looking directly at both the victim and the predator. These contemporary characters connect to Peter and Valerie, who likewise choose to understand their gifts as a step to controlling them. These adaptations denounce the abuse while offering the victim more than blame. Likewise, instead of condemning the predator, they recognize the humanity of the Wolf, giving the good a chance to emerge. These reconsiderations go beyond the gothic dualities of good/evil and human/animal, "juxtaposing potential revolution and possible reaction" (Hogle 2002, p. 13).

## 6. Conclusions

By effectively expanding the gothic qualities embedded in "Little Red Riding Hood", these texts explore issues that are both timeless and timely. The instability of the tale's characters makes it a particularly rich landscape for adaptations to explore gothic themes of hidden natures, as well as the potential for transformation. These figures likewise escape their narrative's confines, to explore 21st-century challenges and ideas. Together, these screen adaptations of "Little Red Riding Hood" frankly explore the vulnerability of pre- and postadolescent girls to sexual predators. Instead of relegating the abuse and recovery to allegory or metaphor, these contemporary adaptations admit the effect of actual attacks

as well as threats, allowing girls' positions more complicated than passive victims or silent survivors. Innocence is not required to be a victim—Vanessa is not more deserving of abuse because she is not "sweet". In addition, Vanessa and Valerie are not confined by their family legacies. Instead, they have the ability to choose their own path, within or beyond the woods.

The supporting characters are equally fruitful to gothic explorations, with their own hidden natures and potential. Like the Mirror Queen, Granny does not need to be weak or postsexual. In contemporary adaptations, the complex natures of the previous victims can now intersect with both the predator and vigilante. The Wolf, whether a werewolf or a charming man, most reflects contemporary concerns, as society has become painfully aware that monsters can be anyone. However, while Perrault was content to blame Red, adaptations more often hold the predators accountable. Not only are they not excused as just following their natures, their natures, like their bodies, are no longer considered destiny—female and male characters alike possess dual natures, with the ability to choose their path.

While my students were at first reluctant to let go of their brightly hued story of a young girl skipping through the woods, they appreciated the richer symbolism and relevance of Perrault's and the Grimms' stories. However, even more, they valued the deeper psychological journeys found in contemporary film and television adaptations. These adaptations may have been particularly relevant to them as young adults, as they were likewise coming to terms with their hidden natures, as well as their own position in society: target, victim, predator, survivor, vigilante, or some combination. Instead of Perrault's judgmental moral, together these gothic revisionings of "Little Red Riding Hood" offer their audiences the hope of redemption through transformation.

**Funding:** This research received no external funding.

**Institutional Review Board Statement:** Not applicable.

**Informed Consent Statement:** Not applicable.

**Data Availability Statement:** No new data were created or analyzed in this study. Data sharing is not applicable to this article.

**Conflicts of Interest:** The author declares no conflict of interest.

## Notes

[1] For simplicity, I will refer to the tale generically as "Little Red Riding Hood", as it is popularly known. This involves conflating Perrault's "Little Red Riding Hood" and the Brothers Grimm's "Little Red Cap". Similarly, I will refer to the generic character as "Red".

[2] *Into the Woods* debuted on stage in San Diego in 1986, opening on Broadway in 1987. Its original Broadway production was followed by a 1988 US national tour, a 1990 West End production, a 1997 tenth anniversary concert, a 2002 Broadway revival, and a 2010 London revival. Major Off-Broadway productions were staged in 2012 at Shakespeare in the Park and 2015 at the Roundabout Theatre. A limited-run second Broadway revival opened in 2022, followed by a national tour. Not surprisingly, the 2014 film attracted an all-star cast.

[3] Lilla Crawford was 13 at the time of the film—so she is both Little yet potentially not.

[4] Beckett (2002) further notes that Dahl's "Little Red Riding Hood" also sports a fur coat as well as a pigskin bag—after she killed the talking animals herself (p. 128).

[5] A Blood Moon is the name given to a total lunar eclipse. In *The Brothers Grimm*, the Mirror Queen also needs a Blood Moon for the ritual to restore their beauty.

[6] In the television series *Once Upon a Time* (2011–2017), the grandmother, mother and Ruby are all part of the werewolf line.

[7] This question of whether monstrosity can be resisted is foundation to the TV series fantasy *Grimm* (2011–2017), with a Wolfman best friends with a demon-hunter—both inherited roles.

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
