# Peer review of "Visions of Red Riding Hood: Transformative Bodies in Contemporary Adaptations"

_humanities, doi:10.3390/h12030048_

Round 1
Reviewer 1 Report
Generally, this is a really strong essay, but the stakes for this overall argument could be developed more in the conclusion.

Very minor typos.
Author Response
Point 1: What feels like it’s missing is a stronger overall sense of the stakes, specifically, why these changes matter. The Conclusions section moves toward this, but it could be developed more clearly. In particular, I’d like to see clearer answers to one or more of
these questions:
o How do these adaptations adjust our overall understanding of the “Little Red
Riding Hood” story and its role in society/culture?
o What do these contemporary texts tell us about the gothic in contemporary
culture?
o What do the changes in these new versions tell us about the process and/or
purpose of adaptation as such?
Response 1: I have responded to the Reviewer's call for a "stronger overall sense of the stakes" with additions and edits to the Introduction, the Conclusion, as well as the ending of my sections. This may be edited as needed.
Specific/Line Comments:
Point 2: Abstract Line 4: Include Perrault’s first name the first time he is mentioned.
Point 2: Corrected
Point 3: Abstract Lines 5-6: The timeframe for “When popular culture turned to fairy tales, reimagining them as gothic tales for adults” isn’t clear. What time frame is this essay looking at?
Point 3: Corrected
Point 4: Abstract Line 11: Cut one instance of the word direct here.
Point 4: Corrected
Point 5: Line 44: Always include an author/scholar’s first name whenever mentioning them for the first time.
Point 5: Checked with Editor; this is correct as is with Chicago; not changed.
Point: 6: Line 72: I think this should be “Grimm-era,” unless this is meant to be a play on grimmer/Grimm. If this is a play on words, I’m definitely here for it.
Point 6: This is a play on words; not changed. I am open to different punctuation to make clearer.
Point 7: Line 103: Close quotation marks are missing; Line 125: There’s a space missing between the period and Having-
Point 7: Corrected
I also reviewed and made some minor corrections suggested by GrammarCheck that I had missed previously.
Thank you.

Reviewer 2 Report
This is a promising article that could be very good with some editing, to bring out the various threads running through it that are currently left dangling. The major issue is that I feel throughout the article that I’ve come in in the middle of a conversation that's already started, suggesting that this was perhaps adapted from something longer where the texts and ideas (especially that of transformation, which repeatedly crops up but is never fully explored) have been outlined more slowly and clearly, and in a more logical order. If this is the case, then some of what has been removed needs to be put back in here, obviously within the limits of the word count, if one applies here.
It would be useful to have a clear argument and structure laid out early on. The introduction to the teaching context is interesting, but could possibly be preceded by something giving a stronger signpost for where the article is going and why. It might also be useful to provide a brief summary of what you see as the “basic” or most familiar version of the plot of the “Red Riding Hood” story, so that all of the elements to be discussed are clearly set out from the beginning. Some of what is said in the conclusion could be moved to the introduction, and doing so would help to guide the argument as it develops.
Specific comments:
· A bit more on adaptation theory, and ideally on girlhood in fiction and popular culture, would be useful to help ground the argument.
· p.2: “the late 20th-century version of the tales connected to new attitudes toward female sexuality and agency – traces of which existed in the Perrault and Grimms’ tales.” This is a little ambiguous and would benefit from a small bit of expansion. What new attitudes, exactly? And what traces can be found in the earlier texts?
· p.2: “violence comes more naturally to these female protagonists, bordering on bloodlust.” This is never fully explored in the article – except briefly in relation to Freeway. Either omit this line, or expand on the point.
· p.2: “British and American rewritings of the tale preceded film adaptations.” Does this mean books of fairy tales for children, like Ladybird books etc? Or something else? Being as specific as possible here will make things clearer for readers who may not be familiar with the history of these retellings. The “Disneyfication” mentioned on p.3 equally needs to be elaborated on – what do you see as the message of these more familiar adaptations? How do they present girlhood in general, and Red in particular?
· A bit more detail on “the generic structure of 18th-century gothic” (p.2) would be useful, if this is possible within the word count.
· It’s important to tease out whether the female protagonist is “sweet” and virginal; sexually precocious and therefore “forward”; or vengeful and violent. It’s a case of spelling out that the various versions vacillate between these options/ extremes. You point out quite nicely that there seems to be a tendency for these texts to wobble between saying that Red is “humble,” ordinary, and innocent, and implying on the other hand that she’s wayward and dangerously loose of morals. This contradiction is central to misogynistic depictions of female characters, so being explicit from the beginning about the fact that it’s there would add substantially to the article. The quotation on p. 5 (“As Bacchilega (1999) notes, ‘devoured or domesticated, charged with sin or in charge of the feminine hearth’”) might be a good way to start the whole essay and to signal early on that the tendency for the tale and its retellings to flip back and forth between these categories is part of the instability discussed in the opening pages.
· The shift into discussing particular texts on p.3 is very abrupt – it would be useful to have a sense of the direction of the article, and which texts will be focused on above others and why. For example, the Angela Carter and Tanith Lee texts are mentioned early on but never discussed – why focused on the ones you examine in detail? What do they do or offer that others don’t? A quick plot summary of The Brothers Grimm would also be useful, so you aren’t discussing intricate details before explaining what’s going on.
· p.6: “The independent film Freeway (Bright, 1996) slightly predates Hollywood’s interest in fairy tales.” When do you date this new/renewed interest to, and why do you think this shift takes place? A paragraph/ section on this before you get into discussing individual texts will help contextualise the argument before you get into specifics.
· p.7: “in adaptations, the protagonist often takes Perrault’s moral too much to heart, closing herself not only to wolves but good men.” What are these adaptations, and what exactly do you mean by this? Expand and explain, or else show this as you go along in the previous pages.
· p.8 – who is Ruby?
Suggested Reading
Nancy Armstrong, Desire and Domestic Fiction (a very useful introduction to gender roles and sexuality in Western culture).
Eugenia C. De LaMotte, Perils of the Night (a very clear and detailed introduction to gender in the gothic).
Tania Modleski, Loving with a Vengeance (might be useful for some of the werewolf vs potential lover issues that are raised here).
Maria Tatar, The Classic Fairy Tales (a useful source of different versions and history of these reimaginings).
This is generally well written, though a thorough proofread, possibly by someone other than the author in consultation with the author, will help catch any minor oddities in style, along with typos and incomplete sentences. Reading the work aloud, ideally to an audience unfamiliar with the texts, or even just to an imagined audience, might also help catch some of the places where phrasing is unclear, or where plot points need to be explained more slowly.
Author Response
Point 1: It would be useful to have a clear argument and structure laid out early on.
Response 1: I have made changes to the Introduction's first and final paragraphs.
Point 2: A bit more on adaptation theory, and ideally on girlhood in fiction and popular culture, would be useful to help ground the argument.
Response 2: Besides connecting to portrayal of abuse with O'Malley, this falls beyond the scope of this essay or special issue, and I am constrained by word count.
Point 3: p.2: “the late 20th-century version of the tales connected to new attitudes toward female sexuality and agency – traces of which existed in the Perrault and Grimms’ tales.” This is a little ambiguous and would benefit from a small bit of expansion. What new attitudes, exactly? And what traces can be found in the earlier texts?
Response 3: Sentence edited
Point 4: p.2: “violence comes more naturally to these female protagonists, bordering on bloodlust.” This is never fully explored in the article– except briefly in relation to Freeway. Either omit this line, or expand on the point.
Response 4: Reviewer missed semi-colon; this comment refers specifically to choices of Carter, Lee and Dahl. No change.
Point 5: · p.2: “British and American rewritings of the tale preceded film adaptations.” Does this mean books of fairy tales for children, like Ladybird books etc? Or something else? Being as specific as possible here will make things clearer for readers who may not be familiar with the history of these retellings.
Response 5:: Edited for clarity; referring to Carter, Lee and Dahl.
Point 6: A bit more detail on “the generic structure of 18th-century gothic” (p.2) would be useful, if this is possible within the word count.
Response 6: Gothic elements most relevant are noted in this paragraph. I am constrained by word count to do more.
Point 7: The shift into discussing particular texts on p.3 is very abrupt – it would be useful to have a sense of the direction of the article, and which texts will be focused on above others and why
Response 7: This is now set up in Introduction.
Point 8: A quick plot summary of The Brothers Grimm would also be useful, so you aren’t discussing intricate details before explaining what’s going on.
Response 8: Added and re-ordered.
Point 9: p.6: “The independent film Freeway (Bright, 1996) slightly predates Hollywood’s interest in fairy tales.” When do you date this new/renewed interest to, and why do you think this shift takes place? A paragraph/ section on this before you get into discussing individual texts will help contextualise the argument before you get into specifics.
Response 9: Dates added in Introduction. Discussion of Hollywood context beyond scope.
Point 10: p.7: “in adaptations, the protagonist often takes Perrault’s moral too much to heart, closing herself not only to wolves but good men.” What are these adaptations, and what exactly do you mean by this? Expand and explain, or else show this as you go along in the previous pages.
Response 10: Sentence edited to make it clear that I am connecting to other adaptations, not examined in this essay.
Point 11: p.8 – who is Ruby?
Response 11: Reference removed.
Reviewer 3 Report
The essay provides a survey of recent adaptations of 'Little Red Riding Hood', particularly with reference to sexual abuse and Red's adolescent sexuality and female sexual agency. The article's chief contribution comes from this discussion of a wide range of recent adaptations of a classic fairy tale. The bibliography offers a combination of adaptation, Gothic and fairy tale criticism.
I think the introduction could be reframed a little to provide a stronger sense of the argument and the materials under discussion. While the author is using recent editions of Perrault and Grimm, the fairy tales could be dated more securely in their original contexts. I would also like to see the adaptations discussed in the essay briefly introduced in the opening paragraphs so their inclusion doesn't come as a surprise later.
The essay is on the whole clearly written and structured. There are occasional minor typos that careful proofreading should eliminate.
Author Response
Response to Reviewer 3 Comments
I think the introduction could be reframed a little to provide a stronger sense of the argument and the materials under discussion.
Point 1: I have made the argument more direct.
While the author is using recent editions of Perrault and Grimm, the fairy tales could be dated more securely in their original contexts.
Point 2: I have added the context of the Grimm and Perrault’s original publication.
I would also like to see the adaptations discussed in the essay briefly introduced in the opening paragraphs so their inclusion doesn't come as a surprise later.
Point 3: All films are now mentioned in Introduction.
The essay is on the whole clearly written and structured. There are occasional minor typos that careful proofreading should eliminate.
Point 4: I have further proofread.